# THE BENEFITS OF OVER-PARAMETERIZATION AT INITIALIZATION IN DEEP RELU NETWORKS

## ABSTRACT

It has been noted in existing literature that over-parameterization in ReLU networks generally improves performance. While there could be several factors involved behind this, we prove some desirable theoretical properties at initialization which may be enjoyed by ReLU networks. Specifically, it is known that He initialization in deep ReLU networks asymptotically preserves variance of activations in the forward pass and variance of gradients in the backward pass for infinitely wide networks, thus preserving the flow of information in both directions. Our paper goes beyond these results and shows novel properties that hold under He initialization: i) the norm of hidden activation of each layer is equal to the norm of the input, and, ii) the norm of weight gradient of each layer is equal to the product of norm of the input vector and the error at output layer. These results are derived using the PAC analysis framework, and hold true for finitely sized datasets such that the width of the ReLU network only needs to be larger than a certain finite lower bound. As we show, this lower bound depends on the depth of the network and the number of samples, and by the virtue of being a lower bound, over-parameterized ReLU networks are endowed with these desirable properties. For the aforementioned hidden activation norm property under He initialization, we further extend our theory and show that this property holds for a finite width network even when the number of data samples is infinite. Thus we overcome several limitations of existing papers, and show new properties of deep ReLU networks at initialization.

## 1 INTRODUCTION

Deep rectifier (ReLU) networks are popular in deep learning due to their ease of training and state-of-the-art generalization. This success of deep rectifier networks can be partly attributed to *good* initialization strategies (for example Glorot & Bengio (2010); He et al. (2015)). Essentially, good parameter initializations guarantee that there is no exploding or vanishing of information across hidden layers. These properties help gradient descent based optimization methods in navigating the complex non-linear loss landscape of deep networks by initializing them at a *good* starting point where training can begin. Such favorable properties promised by these initialization strategies are (in most cases) shown to hold true in asymptotic settings where the network width tends to infinity and/or under strict assumptions made about the distribution of the input data. A detailed account of these existing papers and a contrast between these papers and our work is discussed in section 2.

Our paper relaxes the aforementioned assumptions made in previous papers. Further, we show novel properties that hold for deep ReLU networks at initialization when using the He initialization scheme (He et al., 2015). Specifically, we show that deep ReLU networks obey the following properties in the forward (Eq. 1) back backward (Eq. 2) pass (see section 3 for notations),

$$\|\mathbf{h}^l\|_2 \approx \|\mathbf{x}\|_2 \quad \forall l \in \{1, 2, \cdots, L\} \tag{1}$$

$$\|\frac{\partial \ell(f_\theta(\mathbf{x}), \mathbf{y})}{\partial \mathbf{W}^l}\|_F \approx \|\delta(\mathbf{x}, \mathbf{y})\|_2 \cdot \|\mathbf{x}\|_2 \quad \forall l \in \{1, 2, \cdots, L\} \tag{2}$$

We refer to the above properties as as the the *activation norm equality* and the *gradient norm equality* property.

Further, we derive a finite lower bound on the width of the hidden layers for which the above results hold (i.e., the network needs to be sufficiently over-parameterized) in contrast to a number of previous papers that assume infinitely wide layers.

We do not make any assumption on the data distribution as done in a number of previous papers that study initialization. Further, our results hold for an infinite stream of data for the activation norm equality property and for any finite dataset in the backward pass.

Thus we have relaxed a number of assumptions made in previous research work that focus on deriving initialization strategies for deep ReLU networks. Our results showing the connection between activation norm and input norm (and similarly the property for gradients) for deep ReLU networks can be utilized in further research studies.

## 2    RELATION WITH EXISTING WORK

The seminal work of Glorot & Bengio (2010) studied for the first time a principled way to initialize deep networks to avoid exploding/vanishing gradient problem (EVGP). Their analysis however is done for deep linear networks. The analysis by He et al. (2015) follows the derivation strategy of Glorot & Bengio (2010) except they tailor their derivation for deep ReLU networks. However, both these papers make a strong assumption that the dimensions of the input are statistically independent and that the network width is infinite. Our results do not make these assumptions.

Saxe et al. (2013) introduce the notion of dynamical isometry which is achieved when all the singular values of the input-output Jacobian of the network is 1. They show that deep linear networks achieve dynamical isometry when initialized using orthogonal weights and this property allows fast learning in such networks.

Poole et al. (2016) study how the norm of hidden activations evolve when propagating an input through the network. Pennington et al. (2017; 2018) study the exploding and vanishing gradient problem in deep ReLU networks using tools from free probability theory. Under the assumption of an infinitely wide network, they show that the average squared singular value of the input-output Jacobian for deep ReLU network is 1 when initialized appropriately. Our paper on the other hand shows that deep ReLU networks are norm preserving maps at appropriate initialization. Further, we show there exists a finite lower bound on the width of the network for which the Frobenius norm of the hidden layer-output Jacobian (equivalently the sum of its squared singular values) are equal across all hidden layers.

Hanin & Rolnick (2018) show that for a fixed input, the variance of the squared norm of hidden layer activations are bounded from above and below for deep ReLU networks to be near the squared norm of the input such that the bound depends on the sum of reciprocal of layer widths of the network. Our paper shows a similar result in a PAC bound sense but as an important difference, we show that these results hold even for an infinite stream of data by making the bound depend on the dimensionality of the input.

Hanin (2018) show that sufficiently wide deep ReLU networks with appropriately initialized weights prevent EVGP in the sense that the fluctuation between the elements of the input-output Jacobian matrix of the network is small. This avoids EVGP because a large fluctuation between the elements of the input-output Jacobian implies a large variation in its singular values. Our paper shows that sufficiently wide deep ReLU networks avoid EVGP in the sense that the norm of the gradient for the weights of each layer is roughly equal to a fixed quantity that depends on the input and target.

Over-parameterization in deep networks has previously been shown to have advantages. Neyshabur et al. (2014); Arpit et al. (2017) show empirically that wider networks train faster (number of epochs) and have better generalization performance. From a theoretical view point, Neyshabur et al. (2018) derive a generalization bound for a two layer ReLU network where they show that a wider network has a lower complexity. Lee et al. (2017) show that infinitely wide deep networks act as a Gaussian process. Arora et al. (2018) show that over-parameterization in deep linear networks acts as a conditioning on the gradient leading to faster convergence, although in this case over-parameterization in terms of depth is studied. Our analysis complements this line of work by showing another advantage of over-parameterization in deep ReLU networks.

## 3    THEORETICAL RESULTS

Let $\mathcal{D} = \{\mathbf{x}_i, \mathbf{y}_i\}_{i=1}^N$ be $N$ training sample pairs of inputs vectors $\mathbf{x}_i \in \mathbb{R}^{n_o}$ and target vectors $\mathbf{y}_i^K$ where $\mathbf{x}_i$'s are sampled from a distribution with support $\mathcal{X}$. Define a $L$ layer deep ReLU network $f_\theta(\mathbf{x}) = \mathbf{h}^L$ with the $l^{th}$ hidden layer's activation given by,

$$\mathbf{h}^l := ReLU(\mathbf{a}^l)$$
$$\mathbf{a}^l := \mathbf{W}^l \mathbf{h}^{l-1} + \mathbf{b}^l \quad l \in \{1, 2, \cdots L\} \tag{3}$$

where $\mathbf{h}^l \in \mathbb{R}^{n_l}$ are the hidden activations, $\mathbf{h}^o$ is the input to the network and can be one of the input vectors $\mathbf{x}_i$, $\mathbf{W}^l \in \mathbb{R}^{n_l \times n_{l-1}}$ are the weight matrices, $\mathbf{b} \in \mathbb{R}^{n_l}$ are the bias vectors which are initialized as 0s, $\mathbf{a}^l$ are the pre-activations and $\theta = \{(\mathbf{W}^l, \mathbf{b}^l)\}_{l=1}^L$.

Define a loss on the deep network function for any given training data sample $(\mathbf{x}, \mathbf{y})$ as,

$$\ell(f_\theta(\mathbf{x}), \mathbf{y}) \tag{4}$$

where $\ell(.)$ is any desired loss function. For instance, $\ell(.)$ can be log loss for a classification problem, in which case $f_\theta(\mathbf{x})$ is transformed using a weight matrix to have dimensions equal to the number of classes and the softmax activation is applied to yield class probabilities (i.e., a logistic regression like model on top of $f_\theta(\mathbf{x})$). However for our purpose, we do not need to restrict $\ell(.)$ to a specific choice, we only need it to be differentiable. We will make use of the notation,

$$\delta(\mathbf{x}, \mathbf{y}) := \frac{\partial \ell(f_\theta(\mathbf{x}), \mathbf{y})}{\partial \mathbf{a}_L} \tag{5}$$

We organize our theoretical results as follows. We first derive the activation norm equality property for finite datasets and then extend these results to infinite dataset setting in section 3.1. We then derive the gradient norm equality property for finite datasets in section 3.2. All formal proofs, if not shown in the main text, are available in the appendix.

### 3.1    ACTIVATION NORM EQUALITY

Consider an $L$ layer deep ReLU network and data $\mathbf{x} \in \mathcal{X}$. We show in this section that the norm of hidden layer activation of any layer is roughly equal to the norm of the input at initialization for all $\mathbf{x} \in \mathcal{X}$ if the network weights are initialized appropriately and the network width is sufficiently large but finite. Specifically we show $\forall l \in [L]$ and $\mathbf{x} \in \mathcal{X}$,

$$\|\mathbf{h}^l\|_2 \approx \|\mathbf{x}\|_2 \tag{6}$$

To achieve this goal, we start with a very simple result– in expectation, ReLU transformation in each layer preserves the norm of its corresponding input if the weights are sampled appropriately. Evaluating this expectation also helps determining the scale of the random initialization that leads to norm preservation.

**Lemma 1** *Let* $\mathbf{v} = ReLU(\mathbf{R}\mathbf{u})$, *where* $\mathbf{u} \in \mathbb{R}^n$, $\mathbf{R} \in \mathbb{R}^{m \times n}$. *If* $\mathbf{R}_{ij} \overset{i.i.d.}{\sim} \mathcal{N}(0, \frac{2}{m})$, *then for any fixed vector* $\mathbf{u}$, $\mathbb{E}[\|\mathbf{v}\|^2] = \|\mathbf{u}\|^2$.

The proof for the above lemma involves simply computing the expectation analytically by exploiting the fact that each dimension of the vector $\mathbf{u}$ is a weighted sum of Gaussian random variables. The above result thus shows that for each layer, initializing its weights from an i.i.d. Gaussian distribution with 0 mean and 2/fan-out variance (viz. He initialization (He et al., 2015)) preserves the norm of its input in expectation. We now derive a lower bound on the width of a ReLU layer so that it can preserve the norm of the input for a single fixed input with $\epsilon$ error margin.

**Lemma 2** *Let* $\mathbf{v} = ReLU(\mathbf{R}\mathbf{u})$, *where* $\mathbf{u} \in \mathbb{R}^n$, $\mathbf{R} \in \mathbb{R}^{m \times n}$. *If* $\mathbf{R}_{ij} \overset{i.i.d.}{\sim} \mathcal{N}(0, \frac{2}{m})$, *and* $\epsilon \in [0, 1)$, *then for any fixed vector* $\mathbf{u}$,

$$\Pr\left(|\|\mathbf{v}\|^2 - \|\mathbf{u}\|^2| \le \epsilon \|\mathbf{u}\|^2\right) \ge 1 - 2\exp\left(-m\left(\frac{\epsilon}{4} + \log \frac{2}{1 + \sqrt{1 + \epsilon}}\right)\right) \tag{7}$$

The proof of this lemma involves a direct application of the Chernoff bounding technique. Now we use the above lemma to show that the norm of hidden activations equal the norm of inputs within a specified margin and for a finite size dataset for a deep ReLU network.

**Theorem 1** *Let $\mathcal{D}$ be a fixed dataset with $N$ samples and define a $L$ layer ReLU network $f_\theta(.)$ as shown in Eq. 3 such that each weight matrix $\mathbf{W}^l \in \mathbb{R}^{n_l \times n_{l-1}}$ has its elements sampled as $W_{ij}^l \overset{i.i.d.}{\sim} \mathcal{N}(0, \frac{2}{n_l})$ and biases $\mathbf{b}^l$ are set to zeros. Then for any sample $(\mathbf{x}, \mathbf{y}) \in \mathcal{D}$ and $\epsilon \in [0, 1)$, we have that,*

$$\Pr\left((1 - \epsilon)^L \|\mathbf{x}\|^2 \leq \|f_\theta(\mathbf{x})\|^2 \leq (1 + \epsilon)^L \|\mathbf{x}\|^2\right)$$

$$\geq 1 - \sum_{l'=1}^{L} 2N \exp\left(-n_{l'}\left(\frac{\epsilon}{4} + \log\frac{2}{1 + \sqrt{1 + \epsilon}}\right)\right) \tag{8}$$

While the statement of the above theorem only talks about the norm of the final output of the network, it equally applies to any hidden layer $l$ as well since the theorem can be applied equivalently to a $l$ layer network.

Having proved the activation norm equality property in the finite dataset setting above, we now turn our attention to the case of infinite dataset case. To do so, we first prove a non-trivial result where we use lemma 2 to show how a lower bound on the width of an individual ReLU layer can be computed such that this layer preserves the norm of an infinite stream of inputs.

**Lemma 3** *Let $\mathcal{X}$ be a $d \leq n$ dimensional subspace of $\mathbb{R}^n$ and $\mathbf{R} \in \mathbb{R}^{m \times n}$. If $\mathbf{R}_{ij} \overset{i.i.d.}{\sim} \mathcal{N}(0, \frac{2}{m})$, $\epsilon \in [0, 1)$, and,*

$$m \geq \frac{1}{\epsilon/12 - \log(0.5(1 + \sqrt{1 + \epsilon/3}))} \cdot \left(d\log\frac{2}{\Delta} + \log\frac{4}{\delta}\right) \tag{9}$$

*then with probability at least $1 - \delta$,*

$$(1 - \epsilon)\|\mathbf{u}\|^2 \leq \|ReLU(\mathbf{Ru})\|^2 \leq (1 + \epsilon)\|\mathbf{u}\|^2 \quad \forall \mathbf{u} \in \mathcal{X} \tag{10}$$

*where $\Delta := \min\{\frac{\epsilon}{3\sqrt{d}}, \frac{\sqrt{\epsilon}}{\sqrt{3}d}\}$.*

***Proof Sketch****: The core idea behind the proof is inspired by lemma 10 of Sarlos (2006). Without any loss of generality, we will show the norm preserving property for any unit vector $\mathbf{u}$ in the $d$ dimensional subspace $\mathcal{X}$ of $\mathbb{R}^n$. This is because for any arbitrary length vector $\mathbf{u}$, $\|ReLU(\mathbf{Ru})\| = \|\mathbf{u}\| \cdot \|ReLU(\mathbf{R}\hat{\mathbf{u}})\|$. The idea then is to define a grid of finite points over $\mathcal{X}$ on $[-1, 1]^d$ with interval size depending on $\epsilon$, such that every unit vector $\hat{\mathbf{u}}$ in $\mathcal{X}$ is close enough to one of the grid points. Then, if we choose the width of the layer to be large enough to approximately preserve the length of the finite number of grid points, we can guarantee that the length of any arbitrary unit vector approximately remains preserved as well within the derived margin of error. The formal proof can be found in the appendix.*

We now extend the above lemma to a deep ReLU network and show our main result for the forward pass that the norm of hidden activations equal the norm of input within some distortion margin, for an infinite stream of input data for a sufficiently large (but finite) width deep ReLU network.

**Theorem 2** *Define a $L$ layer ReLU network $f_\theta(.)$ as shown in Eq. 3 such that each weight matrix $\mathbf{W}^l \in \mathbb{R}^{n_l \times n_{l-1}}$ has its elements sampled as $W_{ij}^l \overset{i.i.d.}{\sim} \mathcal{N}(0, \frac{2}{n_l})$ and biases $\mathbf{b}^l$ are set to zeros. Let $\mathcal{X}$ be a $d \leq n$ dimensional subspace of $\mathbb{R}^n$. If $W_{ij}^l \overset{i.i.d.}{\sim} \mathcal{N}(0, \frac{2}{n_l})$, $\epsilon \in [0, 1)$, and,*

$$n_l \geq \frac{1}{\epsilon/12 - \log(0.5(1 + \sqrt{1 + \epsilon/3}))} \cdot \left(d\log\frac{2}{\Delta} + \log\frac{4L}{\delta}\right) \quad \forall l \in [L] \tag{11}$$

*then with probability at least $1 - \delta$,*

$$(1 - \epsilon)^l \|\mathbf{x}\|^2 \leq \|\mathbf{h}^l\|^2 \leq (1 + \epsilon)^l \|\mathbf{x}\|^2 \quad \forall \mathbf{x} \in \mathcal{X} \quad \forall l \in [L] \tag{12}$$

**Proof**: *Since the input lies on a $d$ dimensional subspace of $\mathbb{R}^n$, we apply theorem 3 to the first layer and get the guarantee that the norm of all inputs on the $d$ dimensional subspace are preserved by this layer. Next, we show that since each layer is a linear transform followed by pointwise ReLU non-linearity, and the input takes values in a set defined by the $d$ dimensional subspace of $\mathbb{R}^n$, the output of the first layer will take values in a set that is strictly a subset of a $d$ dimensional subspace. To see this, let $\mathbf{B} \in \mathbb{R}^{n \times d}$ denote a matrix with orthonormal columns describing the basis of the subspace $\mathcal{X}$ on which the input lies, and let $\mathbf{z} \in \mathbb{R}^d$. Then we have that,*

$$\mathbf{x} \in \{\mathbf{Bz}|\mathbf{z} \in \mathbb{R}^d\} \tag{13}$$

*The first layer transforms any input $\mathbf{x}$ as*

$$\mathbf{h}^1 = ReLU(\mathbf{W}^0 \mathbf{x}) \tag{14}$$

$$= ReLU(\mathbf{W}^0 \mathbf{Bz}) \tag{15}$$

*Denote $\mathbf{B}' = \mathbf{W}^0 \mathbf{B}$. Then note that $rank(\mathbf{B}') \leq d$. Let $S_1$ denote the set of values that $\mathbf{h}^1$ can take. Then we have that,*

$$S_1 = \{\mathbf{B}'\mathbf{z}|\mathbf{z} \in \mathbb{R}^d\} \cap \mathbb{R}^{n_1^+} \tag{16}$$

*where $\mathbb{R}^{n^+}$ denote the subset of $\mathbb{R}^n$ where all dimensions take non-negative values. This shows that $\mathbf{h}^1$ takes values in a set that is strictly a subset of a $d$ dimensional subspace of $\mathbb{R}^{n_1}$.*

*Having proved this for first layer, we can recursively apply this strategy to all higher layers since the output of each layer lies on a subset of a subspace. Notice that while doing so, the lower bound on the width of each layer depends only on the subspace dimensionality $d$. Applying union bound over the result of theorem 3 for $L$ layers proves the claim.* $\square$

We note that the lower bound on width derived above depends on two quantities– the depth of the network $L$, and the dimensionality of the subspace $d$ on which the input lies. Specifically, the lower bound on the width becomes larger for larger input dimensionality $d$ and larger network depth $L$ irrespective of the number of data samples, meaning that a wider network is needed as the depth of the network and/or the intrinsic input dimensionality increases.

## 3.2 GRADIENT NORM EQUALITY

Consider any given loss function $\ell(.)$ and a data sample $(\mathbf{x}, \mathbf{y})$, we show in this section that the norm of gradient for the parameter $\mathbf{W}^l$ of the $l^{th}$ layer depends only on the input and output. Specifically, for a wide enough network, the following holds at initialization for all $l \in \{1, 2, \ldots, L\}$ and $\forall \mathbf{x} \in \mathcal{D}$,

$$\|\frac{\partial \ell(f_\theta(\mathbf{x}), \mathbf{y})}{\partial \mathbf{W}^l}\|_F \approx \|\delta(\mathbf{x}, \mathbf{y})\|_2 \cdot \|\mathbf{x}\|_2 \quad \forall l \tag{17}$$

As a first step, we note that the gradient for a parameter $\mathbf{W}^l$ for a sample $(\mathbf{x}, \mathbf{y})$ is given,

$$\frac{\partial \ell(f_\theta(\mathbf{x}), \mathbf{y})}{\partial \mathbf{W}^l} = diag\left(\frac{\partial \ell(f_\theta(\mathbf{x}), \mathbf{y})}{\partial \mathbf{a}^l}\right) \cdot \mathcal{M}_{n_l}(\mathbf{h}^{l-1}) \tag{18}$$

where $\mathcal{M}_{n_l}(\mathbf{h}^{l-1})$ is a matrix of size $n_l \times n_{l-1}$ such that each row is the vector $\mathbf{h}^{l-1}$. Therefore, a simple algebraic manipulation shows that,

$$\|\frac{\partial \ell(f_\theta(\mathbf{x}), \mathbf{y})}{\partial \mathbf{W}^l}\|_F = \|\frac{\partial \ell(f_\theta(\mathbf{x}), \mathbf{y})}{\partial \mathbf{a}^l}\|_2 \cdot \|\mathbf{h}^{l-1}\|_2 \tag{19}$$

In the previous section, we showed that for a sufficiently wide network, $\|\mathbf{h}^l\|_2 \approx \|\mathbf{x}\|_2 \; \forall l$ with high probability. To show that gradient norms of parameters are preserved in the sense shown in Eq. (17), we essentially show that $\|\frac{\partial \ell(f_\theta(\mathbf{x}), \mathbf{y})}{\partial \mathbf{a}^l}\|_2 \approx \|\delta(\mathbf{x}, \mathbf{y})\|_2 \; \forall l$ with high probability for sufficiently wide networks.

Note that $\|\frac{\partial \ell(f_\theta(\mathbf{x}), \mathbf{y})}{\partial \mathbf{a}^L}\|_2 = \|\delta(\mathbf{x}, \mathbf{y})\|_2$ by definition. To show the norm is preserved for all layers, we begin by noting that,

$$\frac{\partial \ell(f_\theta(\mathbf{x}), \mathbf{y})}{\partial \mathbf{a}^l} = \frac{\partial \mathbf{h}^l}{\partial \mathbf{a}^l} \odot \left(\frac{\partial \mathbf{a}^{l+1}}{\partial \mathbf{h}^l}^T \frac{\partial \ell(f_\theta(\mathbf{x}), \mathbf{y})}{\partial \mathbf{a}^{l+1}}\right)$$

$$= \mathbb{1}(\mathbf{a}^l) \odot \left(\mathbf{W}^{l+1^T} \frac{\partial \ell(f_\theta(\mathbf{x}), \mathbf{y})}{\partial \mathbf{a}^{l+1}}\right) \tag{20}$$

where $\odot$ is the point-wise product (or Hadamard product) and $\mathbb{1}(.)$ is the heaviside step function. The following proposition shows that $\mathbb{1}(.)$ follows a Bernoulli distribution w.r.t. the weights given any fixed input at the previous layer.

**Proposition 1** *If network weights are sampled i.i.d. from a Gaussian distribution with mean 0 and biases are 0 at initialization, then conditioned on $\mathbf{h}^{l-1}$, each dimension of $\mathbb{1}(\mathbf{a}^l)$ follows an i.i.d. Bernoulli distribution with probability 0.5 at initialization.*

Given this property of $\mathbb{1}(\mathbf{a}^l)$, we show below that the transformation of type shown in Eq. (20) is norm preserving in expectation.

**Lemma 4** *Let $\mathbf{v} = (\mathbf{R}\mathbf{u}) \odot \mathbf{z}$, where $\mathbf{u} \in \mathbb{R}^n$, $\mathbf{R} \in \mathbb{R}^{m \times n}$ and $\mathbf{z} \in \mathbb{R}^m$. If $\mathbf{R}_{ij} \overset{i.i.d.}{\sim} \mathcal{N}(0, \frac{1}{pm})$ and $\mathbf{z}_i \overset{i.i.d.}{\sim}$ Bernoulli(p), then for any fixed vector $\mathbf{u}$, $\mathbb{E}[\|\mathbf{v}\|^2] = \|\mathbf{u}\|^2$.*

The proof of this lemma involves analytically computing the expectation of the vector norm by exploiting the fact that each dimension of $\mathbf{v}$ is a sum of Gaussian random variables multiplied to an independent Bernoulli random variable. This lemma reveals the variance of the 0 mean Gaussian distribution from which the weights must be sampled in order for the vector norm to be preserved in expectation. Since $\mathbb{1}(\mathbf{a}^l)$ is sampled from a 0.5 probability Bernoulli, we have that the weights must be sampled from a Gaussian with variance $2/m$. We now show this property holds for a finite width network.

**Lemma 5** *Let $\mathbf{v} = (\mathbf{R}\mathbf{u}) \odot \mathbf{z}$, where $\mathbf{u} \in \mathbb{R}^n$, $\mathbf{z} \in \mathbb{R}^m$, and $\mathbf{R} \in \mathbb{R}^{m \times n}$. If $\mathbf{R}_{ij} \overset{i.i.d.}{\sim} \mathcal{N}(0, \frac{1}{0.5m})$, $\mathbf{z}_i \overset{i.i.d.}{\sim}$ Bernoulli(0.5) and $\epsilon \in [0, 1)$, then for any fixed vector $\mathbf{u}$,*

$$\Pr\left(|\|\mathbf{v}\|^2 - \|\mathbf{u}\|^2| \leq \epsilon\|\mathbf{u}\|^2\right)$$
$$\geq 1 - 2\exp\left(-m\left(\frac{\epsilon}{4} + \log\frac{2}{1 + \sqrt{1+\epsilon}}\right)\right) \tag{21}$$

The proof of this lemma involves a direct application of the Chernoff bounding technique. Having shown that a finite width ReLU layer can preserve gradient norm, we now note that we need to apply this result to Eq. (20). In this case, we must substitute the matrix $\mathbf{R}$ in the above lemma with the network's weight matrix $\mathbf{W}^{l+1^T}$. In the previous subsection, we showed that each element of the matrix $\mathbf{W}^{l+1}$ must be sampled from $\mathcal{N}(0, 2/n_{l+1})$ in order for the norm of the input vector to be preserved. However, in order for the Jacobian norm to be preserved, we require $\mathbf{W}^{l+1}$ to be sampled from $\mathcal{N}(0, 2/n_l)$ as per the above lemma. This suggests that if we want the norms to be preserved in the forward and backward pass for a single layer simultaneously, it is beneficial for the width of the network to be close to uniform. The reason we want them to simultaneously hold is because as shown in Eq. (19), in order for the parameter gradient norm to be same for all layers, we need the norm of both the Jacobian $\|\frac{\partial\ell(f_\theta(\mathbf{x}),\mathbf{y})}{\partial\mathbf{a}^l}\|_2$ as well as the hidden activation $\|\mathbf{h}^{l-1}\|_2$ to be preserved throughout the hidden layers. Therefore, assuming the network has a uniform width, we now prove that in deep ReLU networks with He initialization, the norm of weight gradient for each layer is simply a product of norm of the input and norm of the error at output.

**Theorem 3** [1] *Let $\mathcal{D}$ be a fixed dataset with $N$ samples and define a $L$ layer ReLU network as shown in Eq. 3 such that each weight matrix $\mathbf{W}^l \in \mathbb{R}^{n \times n}$ has its elements sampled as $W_{ij}^l \overset{i.i.d.}{\sim} \mathcal{N}(0, \frac{2}{n})$ and biases $\mathbf{b}^l$ are set to zeros. Then for any sample $(\mathbf{x}, \mathbf{y}) \in \mathcal{D}$, $\epsilon \in [0, 1)$, and for all $l \in \{1, 2, \ldots, L\}$ with probability at least,*

$$1 - 4NL\exp\left(-n\left(\frac{\epsilon}{4} + \log\frac{2}{1 + \sqrt{1+\epsilon}}\right)\right) \tag{22}$$

*the following hold true,*

$$(1 - \epsilon)^L\|\mathbf{x}\|^2 \cdot \|\delta(\mathbf{x}, \mathbf{y})\|^2 \leq \|\frac{\partial\ell(f_\theta(\mathbf{x}), \mathbf{y})}{\partial\mathbf{W}^l}\|^2$$
$$\leq (1 + \epsilon)^L\|\mathbf{x}\|^2 \cdot \|\delta(\mathbf{x}, \mathbf{y})\|^2 \tag{23}$$

---

[1]Similar to He et al. (2016), we have assumed that $\frac{\partial\ell(f_\theta(\mathbf{x}),\mathbf{y})}{\partial\mathbf{a}^{l+1}}$ in independent from $\mathbb{1}(\mathbf{a}^l)$ and $\mathbf{W}^{l+1}$ at initialization.

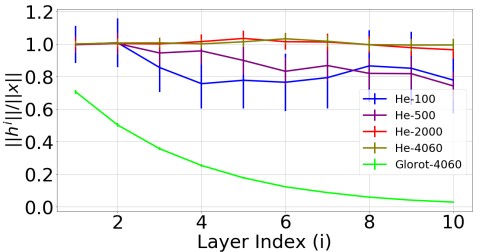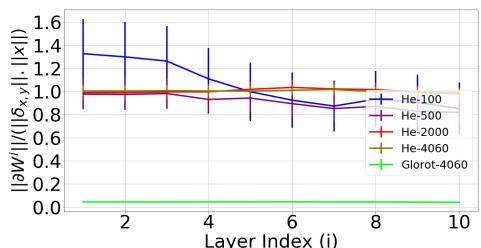

Figure 1: For He initialization, the norm of hidden activation $\mathbf{h}^i$ roughly equals the norm of input $\mathbf{x}$; and the norm of weight gradient $\partial \mathbf{W}^i := \frac{\partial \ell(f_\theta(\mathbf{x}), \mathbf{y})}{\partial \mathbf{W}^i}$ roughly equals the product of norm of input $\mathbf{x}$ and the norm of output error $\delta(\mathbf{x}, \mathbf{y})$ when width is sufficiently large. Glorot initialization does not have this property.

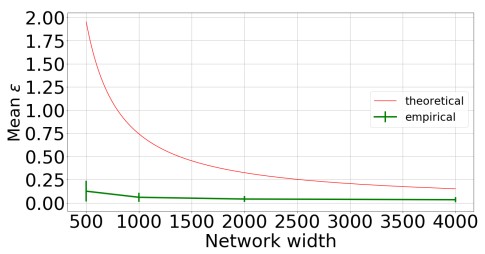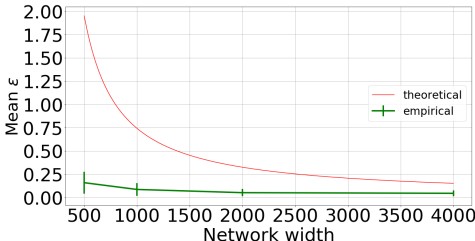

Figure 2: Tightness of lower bounds on network width derived in lemma 2 and lemma 5 shown in the left sub-figure and right sub-figure respectively. See text for more details.

*and*

$$(1 - \epsilon)^l \|\mathbf{x}\|^2 \leq \|\mathbf{h}^l\|^2 \leq (1 + \epsilon)^l \|\mathbf{x}\|^2 \tag{24}$$

We note that even though the theorem relies on the specified independence assumption similar to He et al. (2016), we show that our predictions hold in practice in the next section.

## 4 EMPIRICAL VERIFICATION

### 4.1 NORM PRESERVATION OF ACTIVATION AND GRADIENTS

In this section, we verify the hidden activations have the same norm as input norm $\frac{\|\mathbf{h}^i\|_2}{\|\mathbf{x}\|_2} \approx 1$ (Eq. 6), and the parameter gradient norm approximately equal the product of input norm and output error norm $\frac{\|\frac{\partial \ell(f_\theta(\mathbf{x}), \mathbf{y})}{\partial \mathbf{W}^i}\|_F}{\|\delta(\mathbf{x}, \mathbf{y})\|_2 \cdot \|\mathbf{x}\|_2} \approx 1$ (Eq. 17) for all layer indices $i$ for sufficiently wide deep ReLU networks. For this experiment we choose a 10 layer network with 2000 randomly generated input samples in $\mathbf{R}^{500}$ and randomly generated target labels in $\mathbf{R}^{20}$ and cross-entropy loss. We add a linear layer along with softmax activation to the ReLU network's outputs to make the final output in $\mathbf{R}^{20}$. We use network width from the set $\{100, 500, 2000, 4060\}$. We show results for both He initialization (He et al., 2015) which we theoretically show is optimal, as well as Glorot initialization (Glorot & Bengio, 2010) which is not optimal for deep ReLU nets. As can be seen in figure 1 (left), the mean ratio of hidden activation norm to the input norm over the dataset is roughly 1 with a small standard deviation for He initializaiton. This approximation becomes better with larger width. On the other hand, Glorot initialization fails at preserving activation norm for deep ReLU nets. A similar result can be seen for parameter gradients norms (figure 1 (right)). In the figure we denote $\frac{\partial \ell(f_\theta(\mathbf{x}), \mathbf{y})}{\partial \mathbf{W}^i}$ by $\partial \mathbf{W}^i$. Here we find for He initialization that the norm of weight gradient for each layer is roughly equal to the product of norm of input and norm of error at output, and this approximation becomes stronger for wider networks. Once again Glorot initialization does not have this property.

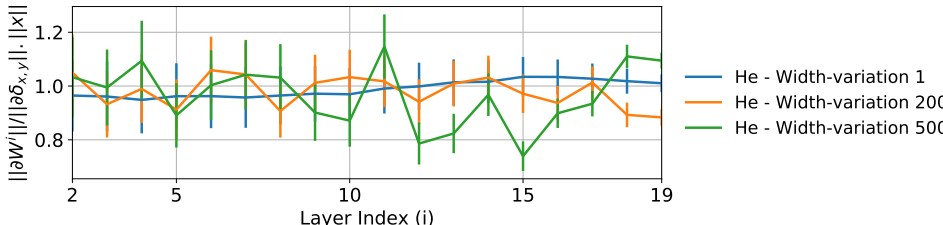

Figure 3: Effect of non-uniformity of width in deep ReLU network on the gradient norm equality property. For each of the three networks, width of each layer is selected independently from $\mathcal{U}(1000 - v, 1000 + v)$, where $v$ is the width variation shown in the plot. Gradient norm equality holds more accurately when width variation is smaller.

## 4.2 TIGHTNESS OF BOUND

In the following experiment we verify the tightness of the bound in lemma 2 (for forward pass) and lemma 5 (for backward pass). To do so, we vary the network width of a one hidden layer ReLU transformation from 500 to 4000, and feed 2000 randomly sampled inputs $\mathbf{x}$ through it. For each sample we measure the distortion $\epsilon$ defined as,

$$\epsilon := \left| 1 - \frac{\|\mathbf{h}\|}{\|\mathbf{x}\|} \right| \tag{25}$$

for the forward pass, and,

$$\epsilon := \left| 1 - \frac{\left\| \frac{\partial \ell(f_\theta(\mathbf{x}), \mathbf{y})}{\partial \mathbf{W}^i} \right\|_F}{\|\delta(\mathbf{x}, \mathbf{y})\|_2 \cdot \|\mathbf{x}\|_2} \right| \tag{26}$$

for the backward pass. Here $\mathbf{h}$ is the output of the one hidden layer ReLU transformation. We compute the mean value of $\epsilon$ for the 2000 examples and plot them against the network width used. We call this the empirical estimate. We simultaneously plot the values of $\epsilon$ predicted by lemma 2 and lemma 5 for failure probability $\delta = 0.05$. We call this the theoretical value. The plot for the forward pass is shown in figure 2 (left). As can be seen, our lower bound on width is an over-estimation but becomes tighter for smaller values of $\epsilon$. A similar result can be seen for lemma 5 in figure 2 (right). Thus our proposed bounds can be improved and we leave that as future work.

## 4.3 EFFECT OF NON-UNIFORMITY OF WIDTH ON GRADIENT NORM EQUALITY

As discussed in section 3.2, gradient norm equality property holds more accurately when deep networks have a more uniform width throughout the layers. To verify this, we construct a 20 layer deep ReLU network such that the width of each layer is determined by independently sampling uniformly between $1000 - v$ and $1000 + v$, where $v$ denotes the amount of width variation chosen for a particular experiment. Once the network architecture is fixed, we initialize the weights with He initialization. We then generate 1000 pairs of input samples and output error similar to the process described in section 4.1 and compute the ratio $\frac{\left\| \frac{\partial \ell(f_\theta(\mathbf{x}), \mathbf{y})}{\partial \mathbf{W}^i} \right\|_F}{\|\delta(\mathbf{x}, \mathbf{y})\|_2 \cdot \|\mathbf{x}\|_2}$. The mean and standard deviation of this value across samples are shown in figure 3 for $v \in \{1, 200, 500\}$. It can be seen that the ratio is closer to 1 with smaller variance when width variation $v$ is small, thus verifying our theoretical prediction.

## 5 CONCLUSION

We derived novel properties that are possessed by deep ReLU networks initialized with He initialization. Specifically, we show that the norm of hidden activations and the norm of weight gradients are a function of the norm of input data and error at output. While deriving these properties, we relaxed most of the assumptions (such as those on input distribution and width of network) made by previous work that study weight initialization in deep ReLU networks. Thus our work establishes that He initialization optimally preserves the flow of information in the forward and backward directions in a stronger setting, and uncovers novel properties.

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

APPENDIX

## A PROOFS

### A.1 PROOFS FOR FORWARD PASS

**Lemma 1** *Let* $\mathbf{v} = ReLU\left(\mathbf{R}\mathbf{u}\right)$*, where* $\mathbf{u} \in \mathbb{R}^n$ *and* $\mathbf{R} \in \mathbb{R}^{m \times n}$*. If* $\mathbf{R}_{ij} \overset{i.i.d.}{\sim} \mathcal{N}(0, \frac{2}{m})$*, then for any fixed vector* $\mathbf{u}$*,* $\mathbb{E}[\|\mathbf{v}\|^2] = \|\mathbf{u}\|^2$*.*

**Proof**: *Define* $a_i = \mathbf{R}_i^T \mathbf{u}$*, where* $\mathbf{R}_i$ *denotes the* $i^{th}$ *row of* $\mathbf{R}$*. Since each element* $\mathbf{R}_{ij}$ *is an independent sample from Gaussian distribution, each* $a_i$ *is essentially a weighted sum of these independent random variables. Thus, each* $a_i \sim \mathcal{N}\left(0, \frac{2}{m}\|\mathbf{u}\|^2\right)$ *and independent from one another. Thus each element* $v_i = ReLU(a_i) \sim \mathcal{N}^R\left(0, \frac{2}{m}\|\mathbf{u}\|^2\right)$ *where* $\mathcal{N}^R$ *denotes the rectified Normal distribution. Our goal is to compute,*

$$\mathbb{E}[\|\mathbf{v}\|^2] = \mathbb{E}[\sum_{i=1}^{m} v_i^2] \tag{27}$$

$$= m\mathbb{E}[v_i^2] \tag{28}$$

*From the definition of* $v_i$*,*

$$\mathbb{E}[v_i] = \frac{1}{2} \cdot 0 + \frac{1}{2}\mathbb{E}[Z] \tag{29}$$

*where* $Z$ *follows a half-Normal distribution corresponding to the Normal distribution* $\mathcal{N}\left(0, \frac{2}{m}\|\mathbf{u}\|^2\right)$*. Thus* $\mathbb{E}[Z] = \sqrt{\frac{2\|\mathbf{u}\|^2}{m}} \cdot \sqrt{\frac{2}{\pi}} = 2\sqrt{\frac{\|\mathbf{u}\|^2}{m\pi}}$*. Similarly,*

$$\mathbb{E}[v_i^2] = 0.5\mathbb{E}[Z^2] \tag{30}$$

$$= 0.5(var(Z) + \mathbb{E}[Z]^2) \tag{31}$$

*Since* $var(Z) = \frac{2}{m}\|\mathbf{u}\|^2(1 - \frac{2}{\pi})$*, we get,*

$$\mathbb{E}[v_i^2] = 0.5\left(\frac{2}{m}\|\mathbf{u}\|^2(1 - \frac{2}{\pi}) + (2\sqrt{\frac{\|\mathbf{u}\|^2}{m\pi}})^2\right) \tag{32}$$

$$= \frac{\|\mathbf{u}\|^2}{m} \tag{33}$$

*Thus,*

$$m\mathbb{E}[v_i^2] = \|\mathbf{u}\|^2 \tag{34}$$

*which proves the claim.* $\square$

**Lemma 2** *Let* $\mathbf{v} = ReLU\left(\mathbf{R}\mathbf{u}\right)$*, where* $\mathbf{u} \in \mathbb{R}^n$*,* $\mathbf{R} \in \mathbb{R}^{m \times n}$*. If* $\mathbf{R}_{ij} \overset{i.i.d.}{\sim} \mathcal{N}(0, \frac{2}{m})$*, and* $\epsilon \in [0, 1)$*, then for any fixed vector* $\mathbf{u}$*,*

$$\Pr\left(|\|\mathbf{v}\|^2 - \|\mathbf{u}\|^2| \le \epsilon\|\mathbf{u}\|^2\right) \ge 1 - 2\exp\left(-m\left(\frac{\epsilon}{4} + \log\frac{2}{1 + \sqrt{1 + \epsilon}}\right)\right) \tag{35}$$

**Proof**: *Define* $\tilde{\mathbf{v}} = \frac{\sqrt{0.5m}}{\|\mathbf{u}\|}\mathbf{v}$*. Then we have that each element* $\tilde{v}_i \sim \mathcal{N}^R\left(0, 1\right)$ *and independent from one another since* $v_i = ReLU(a_i) \sim \mathcal{N}^R\left(0, \frac{2}{m}\|\mathbf{u}\|^2\right)$ *where* $\mathcal{N}^R$ *denotes the rectified Normal distribution. Thus to bound the probability of failure for the R.H.S.,*

$$\Pr\left(\|\mathbf{v}\|^2 \ge (1 + \epsilon)\|\mathbf{u}\|^2\right) = \Pr\left(\frac{\|\mathbf{u}\|^2}{0.5m}\|\tilde{\mathbf{v}}\|^2 \ge (1 + \epsilon)\|\mathbf{u}\|^2\right) \tag{36}$$

$$= \Pr\left(\|\tilde{\mathbf{v}}\|^2 \ge 0.5m(1 + \epsilon)\right) \tag{37}$$

*Using Chernoff's bound, we get for any $\lambda > 0$,*

$$\Pr\left(\|\tilde{\mathbf{v}}\|^2 \geq 0.5m(1+\epsilon)\right) = \Pr\left(\exp(\lambda\|\tilde{\mathbf{v}}\|^2) \geq \exp(\lambda 0.5m(1+\epsilon))\right) \tag{38}$$

$$\leq \frac{\mathbb{E}[\exp(\lambda\|\tilde{\mathbf{v}}\|^2)]}{\exp(0.5m\lambda(1+\epsilon))} \tag{39}$$

$$= \frac{\mathbb{E}[\exp(\sum_{i=1}^m \lambda\tilde{v_i}^2)]}{\exp(0.5m\lambda(1+\epsilon))} \tag{40}$$

$$= \frac{\Pi_{i=1}^m \mathbb{E}[\exp(\lambda\tilde{v_i}^2)]}{\exp(0.5m\lambda(1+\epsilon))} \tag{41}$$

$$= \left(\frac{\mathbb{E}[\exp(\lambda\tilde{v_i}^2)]}{\exp(0.5\lambda(1+\epsilon))}\right)^m \tag{42}$$

*Denote $p(\tilde{v_i})$ as the probability distribution of the rectified Normal random variable $\tilde{v_i}$. Then,*

$$\mathbb{E}[\exp(\lambda\tilde{v_i}^2)] = \int_{-\infty}^{\infty} \exp(\lambda\tilde{v_i}^2)p(\tilde{v_i}) \tag{43}$$

*We know that the mass at $v_i = 0$ is 0.5 and the density between $v_i = 0$ and $v_i = \infty$ follows the Normal distribution. Thus,*

$$\mathbb{E}[\exp(\lambda\tilde{v_i}^2)] = 0.5\exp(0) + \frac{1}{\sqrt{2\pi}}\int_0^{\infty} \exp(\lambda\tilde{v_i}^2 - \tilde{v_i}^2/2) \tag{44}$$

$$= 0.5 + \frac{1}{2\sqrt{(1-2\lambda)}}\frac{\sqrt{2}}{\sqrt{\pi/(1-2\lambda)}}\int_0^{\infty} \exp(-\frac{\tilde{v_i}^2}{2}(1-2\lambda)) \tag{45}$$

*Note that $\int_0^{\infty} \frac{\sqrt{2}}{\sqrt{\pi/(1-2\lambda)}}\int_0^{\infty} \exp(-\frac{\tilde{v_i}^2}{2}(1-2\lambda))$ is the integral of a half Normal distribution corresponding to the Normal distribution $\mathcal{N}(0, 1/(1-2\lambda))$. Thus,*

$$\mathbb{E}[\exp(\lambda\tilde{v_i}^2)] = 0.5 + \frac{1}{2\sqrt{(1-2\lambda)}} \tag{46}$$

*Hence, we get,*

$$\Pr\left(\|\tilde{\mathbf{v}}\|^2 \geq 0.5m(1+\epsilon)\right) \leq \left(0.5\left(1 + \frac{1}{\sqrt{(1-2\lambda)}}\right)\exp(-0.5\lambda(1+\epsilon))\right)^m \tag{47}$$

*The above failure probability can be bounded to be smaller by finding an appropriate value of $\lambda$. We find that $\lambda \approx \frac{0.5\epsilon}{1+\epsilon}$ approximately minimizes the above bound. Substituting this value of $\lambda$ above, we get,*

$$\Pr\left(\|\tilde{\mathbf{v}}\|^2 \geq 0.5m(1+\epsilon)\right) \leq \left(0.5\left(1 + \sqrt{1+\epsilon}\right)\exp(-\frac{\epsilon}{4})\right)^m \tag{48}$$

$$= \exp\left(-m\left(\frac{\epsilon}{4} + \log\frac{2}{1+\sqrt{1+\epsilon}}\right)\right) \tag{49}$$

*Thus,*

$$\Pr\left(\|\mathbf{v}\|^2 \leq (1+\epsilon)\|\mathbf{u}\|^2\right) \geq 1 - \exp\left(-m\left(\frac{\epsilon}{4} + \log\frac{2}{1+\sqrt{1+\epsilon}}\right)\right) \tag{50}$$

*Similarly, to prove the L.H.S. by bounding the probability of failure from the other side,*

$$\Pr\left(\|\mathbf{v}\|^2 \leq (1-\epsilon)\|\mathbf{u}\|^2\right) = \Pr\left(-\|\mathbf{v}\|^2 \geq -(1-\epsilon)\|\mathbf{u}\|^2\right) \tag{51}$$

$$= \Pr\left(-\frac{\|\mathbf{u}\|^2}{0.5m}\|\tilde{\mathbf{v}}\|^2 \geq -(1-\epsilon)\|\mathbf{u}\|^2\right) \tag{52}$$

$$= \Pr\left(-\|\tilde{\mathbf{v}}\|^2 \geq -0.5m(1-\epsilon)\right) \tag{53}$$

*Using Chernoff's bound, we get for any $\lambda > 0$,*

$$\Pr\left(-\|\tilde{\mathbf{v}}\|^2 \geq -0.5m(1-\epsilon)\right) = \Pr\left(\exp(-\lambda\|\tilde{\mathbf{v}}\|^2) \geq \exp(-\lambda 0.5m(1-\epsilon))\right) \tag{54}$$

$$\leq \frac{\mathbb{E}[\exp(-\lambda\|\tilde{\mathbf{v}}\|^2)]}{\exp(-0.5m\lambda(1-\epsilon))} \tag{55}$$

$$= \frac{\mathbb{E}[\exp(-\sum_{i=1}^m \lambda\tilde{v_i}^2)]}{\exp(-0.5m\lambda(1-\epsilon))} \tag{56}$$

$$= \frac{\Pi_{i=1}^m \mathbb{E}[\exp(-\lambda\tilde{v_i}^2)]}{\exp(-0.5m\lambda(1-\epsilon))} \tag{57}$$

$$= \left(\frac{\mathbb{E}[\exp(-\lambda\tilde{v_i}^2)]}{\exp(-0.5\lambda(1-\epsilon))}\right)^m \tag{58}$$

*Performing computations similar to those above to compute the expectation term, we get,*

$$\mathbb{E}[\exp(-\lambda\tilde{v_i}^2)] = 0.5 + \frac{1}{2\sqrt{(1+2\lambda)}} \tag{59}$$

*Hence, we get,*

$$\Pr\left(\|\tilde{\mathbf{v}}\|^2 \leq 0.5m(1-\epsilon)\right) \leq \left(0.5\left(1 + \frac{1}{\sqrt{(1+2\lambda)}}\right)\exp(0.5\lambda(1-\epsilon))\right)^m \tag{60}$$

*Similar to the R.H.S. case, we find that $\lambda \approx \frac{0.5\epsilon}{1-\epsilon}$ approximately minimizes the failure probability,*

$$\Pr\left(\|\tilde{\mathbf{v}}\|^2 \leq 0.5m(1-\epsilon)\right) \leq \left(0.5\left(1 + \sqrt{1-\epsilon}\right)\exp(\frac{\epsilon}{4})\right)^m \tag{61}$$

$$= \exp\left(m\left(\frac{\epsilon}{4} - \log\frac{2}{1+\sqrt{1-\epsilon}}\right)\right) \tag{62}$$

*It can be shown that,*

$$\exp\left(m\left(\frac{\epsilon}{4} - \log\frac{2}{1+\sqrt{1-\epsilon}}\right)\right) \leq \exp\left(-m\left(\frac{\epsilon}{4} + \log\frac{2}{1+\sqrt{1+\epsilon}}\right)\right) \tag{63}$$

*Thus,*

$$\Pr\left(\|\mathbf{v}\|^2 \geq (1-\epsilon)\|\mathbf{u}\|^2\right) \geq 1 - \exp\left(-m\left(\frac{\epsilon}{4} + \log\frac{2}{1+\sqrt{1+\epsilon}}\right)\right) \tag{64}$$

*Using union bound, Eq. (50) and (64) hold together with probability,*

$$\Pr\left((1-\epsilon)\|\mathbf{u}\|^2 \leq \|\mathbf{v}\|^2 \leq (1+\epsilon)\|\mathbf{u}\|^2\right) \geq 1 - 2\exp\left(-m\left(\frac{\epsilon}{4} + \log\frac{2}{1+\sqrt{1+\epsilon}}\right)\right) \tag{65}$$

*This proves the claim.* $\square$

**Theorem 1** *Let $\mathcal{D}$ be a fixed dataset with $N$ samples and define a $L$ layer ReLU network as shown in Eq. 3 such that each weight matrix $\mathbf{W}^l \in \mathbb{R}^{n_l \times n_{l-1}}$ has its elements sampled as $W_{ij}^l \overset{i.i.d.}{\sim} \mathcal{N}(0, \frac{2}{n_l})$ and biases $\mathbf{b}^l$ are set to zeros. Then for any sample $(\mathbf{x}, \mathbf{y}) \in \mathcal{D}$ and $\epsilon \in [0,1)$, we have that,*

$$\Pr\left((1-\epsilon)^L\|\mathbf{x}\|^2 \leq \|f_\theta(\mathbf{x})\|^2 \leq (1+\epsilon)^L\|\mathbf{x}\|^2\right) \geq 1 - \sum_{l'=1}^L 2N\exp\left(-n_{l'}\left(\frac{\epsilon}{4} + \log\frac{2}{1+\sqrt{1+\epsilon}}\right)\right) \tag{66}$$

***Proof****: When feed-forwarding a fixed input through the layers of a deep ReLU network, each hidden layer's activation corresponding to the given input is also fixed because the network is*

*deterministic. Thus applying lemma 2, on each layer's transformation, the following holds true for each $l \in \{1, 2, \cdots L\}$,*

$$\Pr\left((1-\epsilon)\|\mathbf{h}^{l-1}\|^2 \leq \|\mathbf{h}^l\|^2 \leq (1+\epsilon)\|\mathbf{h}^{l-1}\|^2\right) \geq 1 - 2\exp\left(-n_l\left(\frac{\epsilon}{4} + \log\frac{2}{1+\sqrt{1+\epsilon}}\right)\right) \tag{67}$$

*Thus, using union bound, we have the lengths of all the layers until layer $l$ are simultaneously preserved with probability at least,*

$$1 - \sum_{l'=1}^{l} 2\exp\left(-n_{l'}\left(\frac{\epsilon}{4} + \log\frac{2}{1+\sqrt{1+\epsilon}}\right)\right) \tag{68}$$

*Applying union bound again, all the lengths until layer $l$ are preserved simultaneously for $N$ inputs with probability,*

$$1 - \sum_{l'=1}^{l} 2N\exp\left(-n_{l'}\left(\frac{\epsilon}{4} + \log\frac{2}{1+\sqrt{1+\epsilon}}\right)\right) \tag{69}$$

*Finally, we note that the following hold true with the above probability,*

$$(1-\epsilon)\|\mathbf{x}\|^2 \leq \|\mathbf{h}^1\|^2 \leq (1+\epsilon)\|\mathbf{x}\|^2 \tag{70}$$

$$(1-\epsilon)\|\mathbf{h}^1\|^2 \leq \|\mathbf{h}^2\|^2 \leq (1+\epsilon)\|\mathbf{h}^1\|^2 \tag{71}$$

*Substituting $\|\mathbf{h}^1\|^2 \leq (1+\epsilon)\|\mathbf{x}\|^2$ in the R.H.S. of the last equation, and $(1-\epsilon)\|\mathbf{x}\|^2 \leq \|\mathbf{h}^1\|^2$ in the L.H.S. of the last equation, we get,*

$$(1-\epsilon)^2\|\mathbf{x}\|^2 \leq \|\mathbf{h}^2\|^2 \leq (1+\epsilon)^2\|\mathbf{x}\|^2 \tag{72}$$

*Performing substitutions for higher layers similarly yields the claim.* □

**Lemma 3** *Let $\mathcal{X}$ be a $d \leq n$ dimensional subspace of $\mathbb{R}^n$ and $\mathbf{R} \in \mathbb{R}^{m \times n}$. If $\mathbf{R}_{ij} \overset{i.i.d.}{\sim} \mathcal{N}(0, \frac{2}{m})$, $\epsilon \in [0, 1)$, and,*

$$m \geq \frac{1}{\epsilon/12 - \log(0.5(1+\sqrt{1+\epsilon/3}))} \cdot \left(d\log\frac{2}{\Delta} + \log\frac{4}{\delta}\right) \tag{73}$$

*then for all vectors $\mathbf{u} \in \mathcal{X}$, with probability at least $1 - \delta$,*

$$\left| \|ReLU(\mathbf{Ru})\| - \|\mathbf{u}\| \right| \leq \epsilon\|\mathbf{u}\| \tag{74}$$

*where $\Delta := \min\{\frac{\epsilon}{3\sqrt{d}}, \frac{\sqrt{\epsilon}}{\sqrt{3}d}\}$.*

**Proof:** *The core idea behind the proof is inspired by lemma 10 of Sarlos (2006). Without any loss of generality, we will show the norm preserving property for any unit vector $\mathbf{u}$ in the $d$ dimensional subspace $\mathcal{X}$ of $\mathbb{R}^n$. This is because for any arbitrary length vector $\mathbf{u}$, $\|ReLU(\mathbf{Ru})\| = \|\mathbf{u}\| \cdot \|ReLU(\mathbf{R\hat{u}})\|$. The idea then is to define a grid of finite points over $\mathcal{X}$ such that every unit vector $\hat{\mathbf{u}}$ in $\mathcal{X}$ is close enough to one of the grid points. Then, if we choose the width of the layer to be large enough to approximately preserve the length of the finite number of grid points, we essentially guarantee that the length of any arbitrary vector approximately remains preserved.*

*To this end, we define a grid $G$ on $[-1, 1]^d$ with interval of size $\Delta := \min\{\epsilon/\sqrt{d}, \sqrt{\epsilon}/d\}$. Note the number of points on this grid is $\left(\frac{2}{\Delta}\right)^d$. Also, let column vectors of $\mathbf{B} \in \mathbb{R}^{n \times d}$ be the orthonormal basis of $\mathcal{X}$.*

*We now prove the R.H.S. of the bound in the claim. If we consider any unit vector $\hat{\mathbf{u}}$ in $\mathcal{X}$, we can find a point $\mathbf{g}$ on the grid $G$ such that $\|\mathbf{g}\| \leq 1$, and it is closest to $\hat{\mathbf{u}}$ in $\ell^2$ norm, and define $\mathbf{r}' := \hat{\mathbf{u}} - \mathbf{g}$. Thus the vector $\hat{\mathbf{u}}$ can essentially be decomposed as,*

$$\hat{\mathbf{u}} = \mathbf{g} + \mathbf{r}' \tag{75}$$

*Also note that since $\mathbf{r}'$ lies in the span of $\mathcal{X}$, we can represent $\mathbf{r}' := \mathbf{Br}$ for some vector $\mathbf{r}$.*

Now consider the norm of the vector $\hat{\mathbf{u}}$ after the ReLU transformation give by $\|ReLU(\mathbf{R}\hat{\mathbf{u}})\|$. Then we have,

$$\|ReLU(\mathbf{R}\hat{\mathbf{u}})\| = \|ReLU(\mathbf{R}(\mathbf{g} + \mathbf{r}'))\| \tag{76}$$

$$\leq \|ReLU(\mathbf{Rg}) + ReLU(\mathbf{Rr}'))\| \tag{77}$$

$$\leq \|ReLU(\mathbf{Rg})\| + \|ReLU(\mathbf{Rr}'))\| \tag{78}$$

$$\leq \|ReLU(\mathbf{Rg})\| + \|\mathbf{Rr}'\| \tag{79}$$

Similarly, we have,

$$\|ReLU(\mathbf{Rg})\| = \|ReLU(\mathbf{R}(\mathbf{g} + \hat{\mathbf{u}} - \hat{\mathbf{u}}))\| \tag{80}$$

$$\leq \|ReLU(\mathbf{R}\hat{\mathbf{u}}) + ReLU(\mathbf{R}(\mathbf{g} - \hat{\mathbf{u}})))\| \tag{81}$$

$$\leq \|ReLU(\mathbf{R}\hat{\mathbf{u}})\| + \|ReLU(-\mathbf{Rr}'))\| \tag{82}$$

$$\leq \|ReLU(\mathbf{R}\hat{\mathbf{u}})\| + \|\mathbf{Rr}'\| \tag{83}$$

Therefore,

$$\|ReLU(\mathbf{Rg})\| - \|\mathbf{Rr}'\| \leq \|ReLU(\mathbf{R}\hat{\mathbf{u}})\| \leq \|ReLU(\mathbf{Rg})\| + \|\mathbf{Rr}'\| \tag{84}$$

Applying union bound on all the points in G, from lemma 2, we know that with probability at least $1 - \left(\frac{2}{\Delta}\right)^d \exp\left(-m\left(\frac{\epsilon}{4} + \log\frac{2}{1+\sqrt{1+\epsilon}}\right)\right)$,

$$\|ReLU(\mathbf{Rg})\|^2 \leq (1+\epsilon)\|\mathbf{g}\|^2$$

$$\leq 1 + \epsilon \tag{85}$$

$$\leq (1+\epsilon)^2 \tag{86}$$

This can be substituted in the R.H.S. of Eq. (84). Now we only need to upper bound $\|\mathbf{Rr}'\|$. To this end, we rewrite $\|\mathbf{Rr}'\| = \|\mathbf{RBr}\|$. Then,

$$\|\mathbf{RBr}\|^2 = \sum_{i=1}^{d}\sum_{j=1}^{d} < \mathbf{RB}_i r_i, \mathbf{RB}_j r_j > \tag{87}$$

$$\leq 2\sum_{i=1}^{d}\sum_{j=1}^{d} |r_i| \cdot |r_j| \cdot < \frac{1}{\sqrt{2}}\mathbf{RB}_i, \frac{1}{\sqrt{2}}\mathbf{RB}_j > \tag{88}$$

Note that $\frac{1}{\sqrt{2}}\mathbf{R}$ is a matrix whose entries are sampled from $\mathcal{N}(0,1)$. Invoking lemma 6 on the $d^2$ terms in the above sum, we have that with probability at least $1 - 2d^2 \exp\left(-\frac{m}{4}\left(\epsilon^2 - \epsilon^3\right)\right)$,

$$2\sum_{i=1}^{d}\sum_{j=1}^{d} |r_i| \cdot |r_j| \cdot < \frac{1}{\sqrt{2}}\mathbf{RB}_i, \frac{1}{\sqrt{2}}\mathbf{RB}_j > \leq 2\sum_{i=1}^{d}\sum_{j=1}^{d} |r_i| \cdot |r_j| \cdot (< \mathbf{B}_i, \mathbf{B}_j > + \epsilon) \tag{89}$$

$$= 2\sum_{i=1}^{d} r_i^2 \|\mathbf{B}_i\|^2 + 2\sum_{i=1}^{d}\sum_{j=1}^{d} |r_i| \cdot |r_j| \cdot \epsilon \tag{90}$$

$$= 2\|\mathbf{r}\|^2 + 2\epsilon\|\mathbf{r}\|_1^2 \tag{91}$$

Since $\mathbf{r}'$, and hence $\mathbf{r}$ is a point inside one of the grid cells containing the origin, its length can be at most the length of the main diagonal of the grid cell. Formally, $\|\mathbf{r}\| \leq \sqrt{d}\Delta \leq \epsilon$, and $\|\mathbf{r}\|_1 \leq d\Delta \leq \sqrt{\epsilon}$. Subsituting these inequalities in the above equations, we get,

$$\|\mathbf{RBr}\|^2 \leq 4\epsilon^2 \tag{92}$$

Looking back at the R.H.S. of Eq. (84), we have that with probability at least $1 - \left(\frac{2}{\Delta}\right)^d \exp\left(-m\left(\frac{\epsilon}{4} + \log\frac{2}{1+\sqrt{1+\epsilon}}\right)\right) - 2d^2 \exp\left(-\frac{m}{4}\left(\epsilon^2 - \epsilon^3\right)\right)$,

$$\|ReLU(\mathbf{R}\hat{\mathbf{u}})\| \leq 1 + \epsilon + 2\epsilon \tag{93}$$

$$= 1 + 3\epsilon \tag{94}$$

*To prove the L.H.S. of the claimed bound, we can similarly find a point* $\mathbf{g}$ *on the grid* $G$ *such that* $\|\mathbf{g}\| \geq 1$, *and it is closest to* $\hat{\mathbf{u}}$ *in* $\ell^2$ *norm, and define* $\mathbf{r}' := \hat{\mathbf{u}} - \mathbf{g}$. *Then invoking lemma 2, we know that with probability at least* $1 - \left(\frac{2}{\Delta}\right)^d \exp\left(-m\left(\frac{\epsilon}{4} + \log\frac{2}{1+\sqrt{1+\epsilon}}\right)\right)$,

$$\|ReLU(\mathbf{Rg})\|^2 \geq (1 - \epsilon)\|\mathbf{g}\|^2$$
$$\geq 1 - \epsilon \tag{95}$$
$$\geq (1 - \epsilon)^2 \tag{96}$$

*This can be substituted in the L.H.S. of Eq. (84). We then substitute the previously computed upper bound of* $\|\mathbf{RBr}\|^2$ *once again and have that with probability at least* $1 - 2\left(\frac{2}{\Delta}\right)^d \exp\left(-m\left(\frac{\epsilon}{4} + \log\frac{2}{1+\sqrt{1+\epsilon}}\right)\right) - 2d^2 \exp\left(-\frac{m}{4}\left(\epsilon^2 - \epsilon^3\right)\right)$,

$$1 - 3\epsilon \leq \|ReLU(\mathbf{R}\hat{\mathbf{u}})\| \leq 1 + 3\epsilon \tag{97}$$

*Scaling* $\hat{\mathbf{u}}$ *arbitrarily, we equivalently have,*

$$(1 - 3\epsilon)\|\mathbf{u}\| \leq \|ReLU(\mathbf{Ru})\| \leq (1 + 3\epsilon)\|\mathbf{u}\| \tag{98}$$

*Finally, since,*

$$\left(\frac{2}{\Delta}\right)^d \exp\left(-m\left(\frac{\epsilon}{4} + \log\frac{2}{1+\sqrt{1+\epsilon}}\right)\right) \geq d^2 \exp\left(-\frac{m}{4}\left(\epsilon^2 - \epsilon^3\right)\right) \tag{99}$$

*We can further lower bound the success probability of Eq. (98) for mathematical ease as,*

$$1 - 4\left(\frac{2}{\Delta}\right)^d \exp\left(-m\left(\frac{\epsilon}{4} + \log\frac{2}{1+\sqrt{1+\epsilon}}\right)\right) \tag{100}$$

*Therefore to guarantee a success probability of at least* $1 - \delta$, *we bound,*

$$1 - 4\left(\frac{2}{\Delta}\right)^d \exp\left(-m\left(\frac{\epsilon}{4} + \log\frac{2}{1+\sqrt{1+\epsilon}}\right)\right) \geq 1 - \delta \tag{101}$$

*Rearranging the terms in the equality to get a lower bound on* $m$ *and rescaling* $\epsilon$ *proves the claim.* $\square$

## A.2 PROOFS FOR BACKWARD PASS

**Proposition 1** *If network weights are sampled i.i.d. from a Gaussian distribution with mean 0 and biases are 0 at initialization, then conditioned on* $\mathbf{h}^{l-1}$, *each dimension of* $\mathbb{1}(\mathbf{a}^l)$ *follows an i.i.d. Bernoulli distribution with probability 0.5 at initialization.*

**Proof**: *Note that* $\mathbf{a}^l := \mathbf{W}^l \mathbf{h}^{l-1}$ *at initialization (biases are 0) and* $\mathbf{W}^l$ *are sampled i.i.d. from a random distribution with mean 0. Therefore, each dimension* $\mathbf{a}_i^l$ *is simply a weighted sum of i.i.d. zero mean Gaussian, which is also a 0 mean Gaussian random variable.*

*To prove the claim, note that the indicator operator applied on a random variable with 0 mean and symmetric distribution will have equal probability mass on both sides of 0, which is the same as a Bernoulli distributed random variable with probability 0.5. Finally, each dimension of* $\mathbf{a}^l$ *is i.i.d. simply because all the elements of* $\mathbf{W}^l$ *are sampled i.i.d., and hence each dimension of* $\mathbf{a}^l$ *is a weighted sum of a different set of i.i.d. random variables.* $\square$

**Lemma 4** *Let* $\mathbf{v} = (\mathbf{Ru}) \odot \mathbf{z}$, *where* $\mathbf{u} \in \mathbb{R}^n$, $\mathbf{R} \in \mathbb{R}^{m \times n}$ *and* $\mathbf{z} \in \mathbb{R}^m$. *If* $\mathbf{R}_{ij} \overset{i.i.d.}{\sim} \mathcal{N}(0, \frac{1}{pm})$ *and* $\mathbf{z}_i \overset{i.i.d.}{\sim}$ *Bernoulli(p), then for any fixed vector* $\mathbf{u}$, $\mathbb{E}[\|\mathbf{v}\|^2] = \|\mathbf{u}\|^2$.

**Proof**: *Define* $a_i = \mathbf{R}_i^T \mathbf{u}$, *where* $\mathbf{R}_i$ *denotes the* $i^{th}$ *row of* $\mathbf{R}$. *Since each element* $\mathbf{R}_{ij}$ *is an independent sample from Gaussian distribution, each* $a_i$ *is essentially a weighted sum of these independent random variables. Thus, each* $a_i \sim \mathcal{N}\left(0, \frac{1}{pm}\|\mathbf{u}\|^2\right)$ *and independent from one another.*

*Our goal is to compute,*

$$\mathbb{E}[\|\mathbf{v}\|^2] = \sum_{i=1}^{m} \mathbb{E}[(a_i z_i)^2] \tag{102}$$

$$= \sum_{i=1}^{m} \mathbb{E}[a_i^2]\mathbb{E}[z_i^2] \tag{103}$$

$$= m\mathbb{E}[a_i^2]\mathbb{E}[z_i^2] \tag{104}$$

$$= mp(var(a_i) + \mathbb{E}[a_i]^2) \tag{105}$$

$$= \|\mathbf{u}\|^2 \tag{106}$$

*which proves the claim.* □

**Lemma 5** *Let* $\mathbf{v} = (\mathbf{Ru}) \odot \mathbf{z}$*, where* $\mathbf{u} \in \mathbb{R}^n$*,* $\mathbf{z} \in \mathbb{R}^m$*, and* $\mathbf{R} \in \mathbb{R}^{m \times n}$*. If* $\mathbf{R}_{ij} \overset{i.i.d.}{\sim} \mathcal{N}(0, \frac{1}{0.5m})$*,* $\mathbf{z}_i \overset{i.i.d.}{\sim}$ *Bernoulli(0.5) and* $\epsilon \in [0, 1)$*, then for any fixed vector* $\mathbf{u}$*,*

$$\Pr\left(|\|\mathbf{v}\|^2 - \|\mathbf{u}\|^2| \le \epsilon\|\mathbf{u}\|^2\right) \ge 1 - 2\exp\left(-m\left(\frac{\epsilon}{4} + \log\frac{2}{1 + \sqrt{1 + \epsilon}}\right)\right) \tag{107}$$

**Proof**: *Define* $a_i = \mathbf{R}_i^T \mathbf{u}$*, where* $\mathbf{R}_i$ *denotes the* $i^{th}$ *row of* $\mathbf{R}$*. Then, each* $a_i \sim \mathcal{N}\left(0, \frac{1}{0.5m}\|\mathbf{u}\|^2\right)$ *and independent from one another. Define* $\tilde{\mathbf{a}} = \frac{\sqrt{0.5m}}{\|\mathbf{u}\|}\mathbf{a}$*. Then we have that each element* $\tilde{a}_i \sim \mathcal{N}(0, 1)$*.*

*Define* $\tilde{\mathbf{v}}$ *such that* $\tilde{v}_i = \tilde{a}_i z_i$*. Thus to bound the probability of failure for the R.H.S.,*

$$\Pr\left(\|\mathbf{v}\|^2 \ge (1 + \epsilon)\|\mathbf{u}\|^2\right) = \Pr\left(\frac{\|\mathbf{u}\|^2}{0.5m}\|\tilde{\mathbf{v}}\|^2 \ge (1 + \epsilon)\|\mathbf{u}\|^2\right) \tag{108}$$

$$= \Pr\left(\|\tilde{\mathbf{v}}\|^2 \ge 0.5m(1 + \epsilon)\right) \tag{109}$$

*Using Chernoff's bound, we get for any* $\lambda > 0$*,*

$$\Pr\left(\|\tilde{\mathbf{v}}\|^2 \ge 0.5m(1 + \epsilon)\right) = \Pr\left(\exp(\lambda\|\tilde{\mathbf{v}}\|^2) \ge \exp(\lambda 0.5m(1 + \epsilon))\right) \tag{110}$$

$$\le \frac{\mathbb{E}[\exp(\lambda\|\tilde{\mathbf{v}}\|^2)]}{\exp(0.5m\lambda(1 + \epsilon))} \tag{111}$$

$$= \frac{\mathbb{E}[\exp(\sum_{i=1}^{m} \lambda\tilde{v}_i^2)]}{\exp(0.5m\lambda(1 + \epsilon))} \tag{112}$$

$$= \frac{\Pi_{i=1}^m \mathbb{E}[\exp(\lambda\tilde{v}_i^2)]}{\exp(0.5m\lambda(1 + \epsilon))} \tag{113}$$

$$= \left(\frac{\mathbb{E}[\exp(\lambda\tilde{v}_i^2)]}{\exp(0.5\lambda(1 + \epsilon))}\right)^m \tag{114}$$

*Denote* $p(\tilde{a}_i)$ *and* $p(z_i)$ *as the probability distribution of the random variables* $\tilde{a}_i$ *and* $z_i$ *respectively. Then,*

$$\mathbb{E}[\exp(\lambda\tilde{v}_i^2)] = \sum_{z_i} p(z_i) \int_{\tilde{a}_i} p(\tilde{a}_i) \exp(\lambda\tilde{a}_i^2 z_i^2) \tag{115}$$

*Substituting* $p(\tilde{a}_i)$ *with a standard Normal distribution, we get,*

$$\mathbb{E}[\exp(\lambda\tilde{v}_i^2)] = \sum_{z_i} p(z_i) \int_{\tilde{a}_i} \frac{1}{\sqrt{2\pi}} \exp(\lambda\tilde{a}_i^2 z_i^2 - \frac{\tilde{a}_i^2}{2}) \tag{116}$$

$$= \sum_{z_i} p(z_i) \int_{\tilde{a}_i} \frac{1}{\sqrt{2\pi}} \exp(-\frac{\tilde{a}_i^2}{2}(1 - 2\lambda z_i^2)) \tag{117}$$

$$= \sum_{z_i} p(z_i) \int_{\tilde{a}_i} \frac{1}{\sqrt{2\pi}} \cdot \frac{\sqrt{1 - 2\lambda z_i^2}}{\sqrt{1 - 2\lambda z_i^2}} \exp(-\frac{\tilde{a}_i^2}{2}(1 - 2\lambda z_i^2)) \tag{118}$$

$$= \sum_{z_i} p(z_i) \cdot \frac{1}{\sqrt{1 - 2\lambda z_i^2}} \tag{119}$$

*where the last equality holds because the integral of a Gaussian distribution over its domain is 1. Finally, summing over the Bernoulli random variable $z_i$, we get,*

$$\mathbb{E}[\exp(\lambda \tilde{v_i}^2)] = (1 - 0.5) + \frac{1}{\sqrt{1 - 2\lambda}} \tag{120}$$

*Hence, we get,*

$$\Pr\left(\|\tilde{\mathbf{v}}\|^2 \geq 0.5m(1 + \epsilon)\right) \leq \left(0.5\left(1 + \frac{0.5}{\sqrt{(1 - 2\lambda)}}\right)\exp(-0.5\lambda(1 + \epsilon))\right)^m \tag{121}$$

$$\leq \left(0.5\left(1 + \frac{1}{\sqrt{(1 - 2\lambda)}}\right)\exp(-0.5\lambda(1 + \epsilon))\right)^m \tag{122}$$

*We find that the above inequality is identical to that in Eq. (47). Thus $\lambda \approx \frac{0.5\epsilon}{1+\epsilon}$ approximately minimizes the above bound as before. Substituting this value of $\lambda$ above, we get,*

$$\Pr\left(\|\tilde{\mathbf{v}}\|^2 \geq 0.5m(1 + \epsilon)\right) \leq \left(0.5\left(1 + \sqrt{1 + \epsilon}\right)\exp(-\frac{\epsilon}{4})\right)^m \tag{123}$$

$$= \exp\left(-m\left(\frac{\epsilon}{4} + \log\frac{2}{1 + \sqrt{1 + \epsilon}}\right)\right) \tag{124}$$

*Thus,*

$$\Pr\left(\|\mathbf{v}\|^2 \leq (1 + \epsilon)\|\mathbf{u}\|^2\right) \geq 1 - \exp\left(-m\left(\frac{\epsilon}{4} + \log\frac{2}{1 + \sqrt{1 + \epsilon}}\right)\right) \tag{125}$$

*Similarly, to prove the L.H.S. by bounding the probability of failure from the other side,*

$$\Pr\left(\|\mathbf{v}\|^2 \leq (1 - \epsilon)\|\mathbf{u}\|^2\right) = \Pr\left(-\|\mathbf{v}\|^2 \geq -(1 - \epsilon)\|\mathbf{u}\|^2\right) \tag{126}$$

$$= \Pr\left(-\frac{\|\mathbf{u}\|^2}{0.5m}\|\tilde{\mathbf{v}}\|^2 \geq -(1 - \epsilon)\|\mathbf{u}\|^2\right) \tag{127}$$

$$= \Pr\left(-\|\tilde{\mathbf{v}}\|^2 \geq -0.5m(1 - \epsilon)\right) \tag{128}$$

*Using Chernoff's bound, we get for any $\lambda > 0$,*

$$\Pr\left(-\|\tilde{\mathbf{v}}\|^2 \geq -0.5m(1 - \epsilon)\right) = \Pr\left(\exp(-\lambda\|\tilde{\mathbf{v}}\|^2) \geq \exp(-\lambda 0.5m(1 - \epsilon))\right) \tag{129}$$

$$\leq \frac{\mathbb{E}[\exp(-\lambda\|\tilde{\mathbf{v}}\|^2)]}{\exp(-0.5m\lambda(1 - \epsilon))} \tag{130}$$

$$= \frac{\mathbb{E}[\exp(-\sum_{i=1}^{m}\lambda\tilde{v_i}^2)]}{\exp(-0.5m\lambda(1 - \epsilon))} \tag{131}$$

$$= \frac{\Pi_{i=1}^{m}\mathbb{E}[\exp(-\lambda\tilde{v_i}^2)]}{\exp(-0.5m\lambda(1 - \epsilon))} \tag{132}$$

$$= \left(\frac{\mathbb{E}[\exp(-\lambda\tilde{v_i}^2)]}{\exp(-0.5\lambda(1 - \epsilon))}\right)^m \tag{133}$$

*Performing computations similar to those above to compute the expectation term, we get,*

$$\mathbb{E}[\exp(-\lambda\tilde{v_i}^2)] = 0.5 + \frac{1}{\sqrt{(1 + 2\lambda)}} \tag{134}$$

*Hence, we get,*

$$\Pr\left(\|\tilde{\mathbf{v}}\|^2 \leq 0.5m(1 - \epsilon)\right) \leq \left(0.5\left(1 + \frac{0.5}{\sqrt{(1 + 2\lambda)}}\right)\exp(0.5\lambda(1 - \epsilon))\right)^m \tag{135}$$

$$\leq \left(0.5\left(1 + \frac{1}{\sqrt{(1 + 2\lambda)}}\right)\exp(0.5\lambda(1 - \epsilon))\right)^m \tag{136}$$

Similar to the R.H.S. case, we find that $\lambda \approx \frac{0.5\epsilon}{1-\epsilon}$ approximately minimizes the failure probability,

$$\Pr\left(\|\tilde{\mathbf{v}}\|^2 \leq 0.5m(1-\epsilon)\right) \leq \left(0.5\left(1 + \sqrt{1-\epsilon}\right)\exp(\frac{\epsilon}{4})\right)^m \tag{137}$$

$$= \exp\left(m\left(\frac{\epsilon}{4} - \log\frac{2}{1+\sqrt{1-\epsilon}}\right)\right) \tag{138}$$

It can be shown that,

$$\exp\left(m\left(\frac{\epsilon}{4} - \log\frac{2}{1+\sqrt{1-\epsilon}}\right)\right) \leq \exp\left(-m\left(\frac{\epsilon}{4} + \log\frac{2}{1+\sqrt{1+\epsilon}}\right)\right) \tag{139}$$

Thus,

$$\Pr\left(\|\mathbf{v}\|^2 \geq (1-\epsilon)\|\mathbf{u}\|^2\right) \geq 1 - \exp\left(-m\left(\frac{\epsilon}{4} + \log\frac{2}{1+\sqrt{1+\epsilon}}\right)\right) \tag{140}$$

Using union bound, Eq. (125) and (140) hold together with probability,

$$\Pr\left((1-\epsilon)\|\mathbf{u}\|^2 \leq \|\mathbf{v}\|^2 \leq (1+\epsilon)\|\mathbf{u}\|^2\right) \geq 1 - 2\exp\left(-m\left(\frac{\epsilon}{4} + \log\frac{2}{1+\sqrt{1+\epsilon}}\right)\right) \tag{141}$$

This proves the claim. □

**Theorem 2** *Let $\mathcal{D}$ be a fixed dataset with $N$ samples and define a $L$ layer ReLU network as shown in Eq. 3 such that each weight matrix $\mathbf{W}^l \in \mathbb{R}^{n \times n}$ has its elements sampled as $W_{ij}^l \overset{i.i.d.}{\sim} \mathcal{N}(0, \frac{2}{n})$ and biases $\mathbf{b}^l$ are set to zeros. Then for any sample $(\mathbf{x}, \mathbf{y}) \in \mathcal{D}$, $\epsilon \in [0, 1)$, and for all $l \in \{1, 2, \ldots, L\}$ with probability at least,*

$$1 - 4NL\exp\left(-n\left(\frac{\epsilon}{4} + \log\frac{2}{1+\sqrt{1+\epsilon}}\right)\right) \tag{142}$$

*the following hold true,*

$$(1-\epsilon)^L\|\mathbf{x}\|^2 \cdot \|\delta(\mathbf{x}, \mathbf{y})\|^2 \leq \|\frac{\partial\ell(f_\theta(\mathbf{x}), \mathbf{y})}{\partial\mathbf{W}^l}\|^2 \leq (1+\epsilon)^L\|\mathbf{x}\|^2 \cdot \|\delta(\mathbf{x}, \mathbf{y})\|^2 \tag{143}$$

*and*

$$(1-\epsilon)^l\|\mathbf{x}\|^2 \leq \|\mathbf{h}^l\|^2 \leq (1+\epsilon)^l\|\mathbf{x}\|^2 \tag{144}$$

***Proof:*** *From theorem 1, we know that the following holds for all l,*

$$\Pr\left((1-\epsilon)^l\|\mathbf{x}\|^2 \leq \|\mathbf{h}^l\|^2 \leq (1+\epsilon)^l\|\mathbf{x}\|^2\right) \geq 1 - 2NL\exp\left(-n\left(\frac{\epsilon}{4} + \log\frac{2}{1+\sqrt{1+\epsilon}}\right)\right) \tag{145}$$

*On the other hand, we have that,*

$$\frac{\partial\ell(f_\theta(\mathbf{x}), \mathbf{y})}{\partial\mathbf{a}^{L-1}} = \mathbb{1}(\mathbf{a}^{L-1}) \odot \left(\mathbf{W}^{L^T}\delta(\mathbf{x}, \mathbf{y})\right) \tag{146}$$

*From proposition 1, we know that each element of $\mathbb{1}(\mathbf{a}^{L-1})$ follows a Bernoulli distribution with probability 0.5. Thus applying lemma 5 to the above equation (under the assumption that $\delta(\mathbf{x}, \mathbf{y})$ and $\mathbb{1}(\mathbf{a}^l)$ are statistically independent), the following holds for a fixed data sample $(\mathbf{x}, \mathbf{y})$,*

$$\Pr\left((1-\epsilon)\|\delta(\mathbf{x}, \mathbf{y})\|^2 \leq \|\frac{\partial\ell(f_\theta(\mathbf{x}), \mathbf{y})}{\partial\mathbf{a}^{L-1}}\|^2 \leq (1+\epsilon)\|\delta(\mathbf{x}, \mathbf{y})\|^2\right) \geq 1 - 2\exp\left(-n\left(\frac{\epsilon}{4} + \log\frac{2}{1+\sqrt{1+\epsilon}}\right)\right) \tag{147}$$

*Applying union bound on $N$ fixed samples, the following holds for all $N$ samples,*

$$\Pr\left((1-\epsilon)\|\delta(\mathbf{x}, \mathbf{y})\|^2 \leq \|\frac{\partial\ell(f_\theta(\mathbf{x}), \mathbf{y})}{\partial\mathbf{a}^{L-1}}\|^2 \leq (1+\epsilon)\|\delta(\mathbf{x}, \mathbf{y})\|^2\right) \geq 1 - 2N\exp\left(-n\left(\frac{\epsilon}{4} + \log\frac{2}{1+\sqrt{1+\epsilon}}\right)\right) \tag{148}$$

*Similarly,*

$$\Pr\left((1-\epsilon)\|\frac{\partial\ell(f_\theta(\mathbf{x}),\mathbf{y})}{\partial\mathbf{a}^{L-1}}\|^2 \le \|\frac{\partial\ell(f_\theta(\mathbf{x}),\mathbf{y})}{\partial\mathbf{a}^{L-2}}\|^2 \le (1+\epsilon)\|\frac{\partial\ell(f_\theta(\mathbf{x}),\mathbf{y})}{\partial\mathbf{a}^{L-1}}\|^2\right) \ge 1 - 2N\exp\left(-n\left(\frac{\epsilon}{4}+\log\frac{2}{1+\sqrt{1+\epsilon}}\right)\right)$$
(149)

*Combining the the above two results and applying union bound, we get,*

$$\Pr\left((1-\epsilon)^2\|\delta(\mathbf{x},\mathbf{y})\|^2 \le \|\frac{\partial\ell(f_\theta(\mathbf{x}),\mathbf{y})}{\partial\mathbf{a}^{L-2}}\|^2 \le (1+\epsilon)^2\|\delta(\mathbf{x},\mathbf{y})\|^2\right) \ge 1 - 4N\exp\left(-n\left(\frac{\epsilon}{4}+\log\frac{2}{1+\sqrt{1+\epsilon}}\right)\right)$$
(150)

*Extending this to all $L$ layers, we have for all $l$ that,*

$$\Pr\left((1-\epsilon)^{L-l}\|\delta(\mathbf{x},\mathbf{y})\|^2 \le \|\frac{\partial\ell(f_\theta(\mathbf{x}),\mathbf{y})}{\partial\mathbf{a}^l}\|^2 \le (1+\epsilon)^{L-l}\|\delta(\mathbf{x},\mathbf{y})\|^2\right) \ge 1 - 2NL\exp\left(-n\left(\frac{\epsilon}{4}+\log\frac{2}{1+\sqrt{1+\epsilon}}\right)\right)$$
(151)

*Combining the above result with Eq. (145) using union bound, we get for all $l$,*

$$\Pr\left((1-\epsilon)^{L-1}\|\delta(\mathbf{x},\mathbf{y})\|^2\|\mathbf{x}\|^2 \le \|\frac{\partial\ell(f_\theta(\mathbf{x}),\mathbf{y})}{\partial\mathbf{a}^l}\|^2\|\mathbf{h}^{l-1}\|^2 \le (1+\epsilon)^{L-1}\|\delta(\mathbf{x},\mathbf{y})\|^2\|\mathbf{x}\|^2\right)$$

$$\ge 1 - 4NL\exp\left(-n\left(\frac{\epsilon}{4}+\log\frac{2}{1+\sqrt{1+\epsilon}}\right)\right)$$
(152)

*Since,*

$$\|\frac{\partial\ell(f_\theta(\mathbf{x}),\mathbf{y})}{\partial\mathbf{W}^l}\|_2 = \|\frac{\partial\ell(f_\theta(\mathbf{x}),\mathbf{y})}{\partial\mathbf{a}^l}\|_2 \cdot \|\mathbf{h}^{l-1}\|_2 \quad \forall l$$
(153)

*we have proved the claim.* $\square$

**Lemma 6** *(Corollary 2.1 of Kakade & Shakhnarovich (2009)) Let $\mathbf{u_1}, \mathbf{u_2} \in \mathbb{R}^n$ be any two fixed vectors such that $\|\mathbf{u}_1\| \le 1$ and $\|\mathbf{u}_2\| \le 1$, $\mathbf{R} \in \mathbb{R}^{m\times n}$ be a projection matrix where each element of $R$ is drawn i.i.d. from a standard Gaussian distribution, $R_{ij} \sim \mathcal{N}(0,\frac{1}{m})$ and any $\epsilon \in (0,1/2)$*

$$\Pr\left(|<\mathbf{Ru}_1,\mathbf{Ru}_2> - <\mathbf{u}_1,\mathbf{u}_2>| \le \epsilon\right)$$

$$\ge 1 - 4\exp\left(-\frac{m}{4}\left(\epsilon^2-\epsilon^3\right)\right)$$
(154)

**Lemma 7** *Let $\mathbf{v}_1 = (\mathbf{Ru}_1)\odot\mathbf{z}$ and $\mathbf{v}_2 = (\mathbf{Ru}_2)\odot\mathbf{z}$, where $\mathbf{u}_1, \mathbf{u}_2 \in \mathbb{R}^n$, $\mathbf{z} \in \mathbb{R}^m$, and $\mathbf{R} \in \mathbb{R}^{m\times n}$. If $\mathbf{R}_{ij} \overset{i.i.d.}{\sim} \mathcal{N}(0,\frac{1}{0.5m})$, $\mathbf{z}_i \overset{i.i.d.}{\sim}$ Bernoulli(0.5) and $\epsilon \in [0,1)$, then for any fixed vectors $\mathbf{u}_1$ and $\mathbf{u}_2$ s.t. $\|\mathbf{u}_1\| \le 1$ and $\|\mathbf{u}_2\| \le 1$,*

$$\Pr\left(|<\mathbf{v}_1,\mathbf{v}_2> - <\mathbf{u}_1,\mathbf{u}_2>| \le \epsilon\right) \ge 1 - 4\exp\left(-m\left(\frac{\epsilon}{4}+\log\frac{2}{1+\sqrt{1+\epsilon}}\right)\right)$$
(155)

**Proof:** *Applying lemma 5 to vectors $\mathbf{u}_1 + \mathbf{u}_2$ and $\mathbf{u}_1 - \mathbf{u}_2$, we have with probability at least $1 - 4\exp\left(-m\left(\frac{\epsilon}{4}+\log\frac{2}{1+\sqrt{1+\epsilon}}\right)\right)$,*

$$(1-\epsilon)\cdot\|\mathbf{u}_1+\mathbf{u}_2\|^2 \le \|\mathbf{z}\odot\mathbf{Ru}_1 + \mathbf{z}\odot\mathbf{Ru}_2\|^2 \le (1+\epsilon)\cdot\|\mathbf{u}_1+\mathbf{u}_2\|^2 \tag{156}$$

$$(1-\epsilon)\cdot\|\mathbf{u}_1-\mathbf{u}_2\|^2 \le \|\mathbf{z}\odot\mathbf{Ru}_1 - \mathbf{z}\odot\mathbf{Ru}_2\|^2 \le (1+\epsilon)\cdot\|\mathbf{u}_1-\mathbf{u}_2\|^2 \tag{157}$$

*Then notice,*

$$4<\mathbf{v}_1,\mathbf{v}_2> = 4<\mathbf{z}\odot\mathbf{Ru}_1,\mathbf{z}\odot\mathbf{Ru}_2> \tag{158}$$

$$= \|\mathbf{z}\odot\mathbf{Ru}_1 + \mathbf{z}\odot\mathbf{Ru}_2\|^2 - \|\mathbf{z}\odot\mathbf{Ru}_1 - \mathbf{z}\odot\mathbf{Ru}_2\|^2 \tag{159}$$

$$\ge (1-\epsilon)\cdot\|\mathbf{u}_1+\mathbf{u}_2\|^2 - (1+\epsilon)\cdot\|\mathbf{u}_1-\mathbf{u}_2\|^2 \tag{160}$$

$$= 4\cdot<\mathbf{u}_1,\mathbf{u}_2> -2\epsilon\cdot(\|\mathbf{u}_1\|^2 + \|\mathbf{u}_2\|^2) \tag{161}$$

$$\ge 4\cdot<\mathbf{u}_1,\mathbf{u}_2> -4\epsilon \tag{162}$$

*Equivalently,*

$$\cdot <\mathbf{u}_1,\mathbf{u}_2> - <\mathbf{v}_1,\mathbf{v}_2> \le \epsilon \tag{163}$$

*The other side of the claim can be proved similarly.*

