# OpenReview forum: "The Benefits of Over-parameterization at Initialization in Deep ReLU Networks"
_ICLR.cc/2020/Conference — Reject_

### Official Review · AnonReviewer1 · 2019-10-23
**Official Blind Review #1**

**Rating:** 1

**Review:**

The paper studies the norm of hidden activation of each layer and the norm of weight gradient of each layer for deep ReLU neural network. By using concentralization property of the random initialization, the paper derives their expected values and high probability range when the network width is sufficiently wide. The results are correct and the paper is easy to follow. However, the result has been given in previous work. I do not recommend the acceptance.

The result presented in this paper has been covered by a recent work [1]. Please refer to Section 7.1 for the forward part and Section 7.3 for the backward part.

[1] Zeyuan Allen-Zhu, Yuanzhi Li and Zhao Song. A Convergence Theory for Deep Learning via Over-Parameterization

**Experience Assessment:**

I have published one or two papers in this area.

**Review Assessment: Checking Correctness Of Derivations And Theory:**

I carefully checked the derivations and theory.

**Review Assessment: Checking Correctness Of Experiments:**

I assessed the sensibility of the experiments.

**Review Assessment: Thoroughness In Paper Reading:**

I read the paper thoroughly.

---

### Official Review · AnonReviewer2 · 2019-10-23
**Official Blind Review #2**

**Rating:** 3

**Review:**

The paper shows that under He initialization and for sufficiently wide network, (1) the norm of the activations of a L layered ReLU is preserved w.r.t the input across layers and (2) the norm of a weight matrix gradient at different layer is only dependent on the norm of the top-layer error and the input, because the norm of back-propagated gradient is approximately preserved.

The paper is clearly written and easy to read and the proofs are quite straightforward. That being said, the results are not surprising and from my point of view, the overall novelty of this paper is a bit marginal for top-tier conference like ICLR.

Thus I vote for rejection.

**Experience Assessment:**

I have published in this field for several years.

**Review Assessment: Checking Correctness Of Derivations And Theory:**

I assessed the sensibility of the derivations and theory.

**Review Assessment: Checking Correctness Of Experiments:**

I assessed the sensibility of the experiments.

**Review Assessment: Thoroughness In Paper Reading:**

I read the paper at least twice and used my best judgement in assessing the paper.

---

### Official Review · AnonReviewer3 · 2019-10-28
**Official Blind Review #3**

**Rating:** 3

**Review:**

This paper studies initialization techniques for deep ReLU networks from a theoretical standpoint and derives finite layer width concentration bounds to show that with the He initialization scheme, deep ReLU networks preserve the norm of the input sample during a forward pass and the norm of the gradient with respect to the output during a backward pass. The concentration bounds also suggest lower bounds on the width of the ReLU layers. The authors verify their theory with experiments on synthetic data.

While I believe the finite sample concentration bounds for networks initialized using the He initialization are valuable, it seems to me that this work is incremental in terms of understanding the He initialization. The techniques used in the paper are also fairly well known Chernoff bounding techniques and concentration of Gaussian random vectors and matrices.

The authors claims on explaining overparameterization are also overstated in my opinion. While the authors are able to obtain a lower bound on layer widths in order to preserve norms during forward/backward passes at initialization, the lower bound is only dependent on the input dimension and not the size of the dataset, which is the relevant quantity to decide whether a model is under/over parameterized. Even in the authors' bounds for finite datasets, what I can surmise from their results (it would be better to explicitly state it if that is one of the goals of this paper) is that the width of each layer needs to be atleast log(N) where N is the size of the dataset. This is hardly overparameterized.

Furthermore, the authors do not explain how studying the properties of the initialization might help understand generalization at minima. Since gradient descent based techniques seem to prefer solutions that are close to initialization, the analysis in this paper might be a useful starting point in understanding generalization.

The authors could also consider how adding BatchNorm layers and/or Residual connections affect the He initialization scheme, and whether initialization matters for those techniques. Finite width concentration bounds for initialization of networks using Batchnorm/Residual connections could be useful.

To summarize, I do not see how the authors claims about explaining overparameterization (even at initialization) can be made. Without those claims the contribution of this paper is incremental and does not warrant publication at this time. I am willing to adjust my score if the claims about overparameterization are stated explicitly and make sense.

**Experience Assessment:**

I have read many papers in this area.

**Review Assessment: Checking Correctness Of Derivations And Theory:**

I assessed the sensibility of the derivations and theory.

**Review Assessment: Checking Correctness Of Experiments:**

I assessed the sensibility of the experiments.

**Review Assessment: Thoroughness In Paper Reading:**

I read the paper at least twice and used my best judgement in assessing the paper.

---

### Official Review · AnonReviewer4 · 2019-10-30
**Official Blind Review #4**

**Rating:** 3

**Review:**

This work considers random parameter initialization in neural networks (In particular the initialization presented in He et al.) and develops non-asymptotic bounds for the norms and gradients of neural networks during initialization. The authors show that the norms of the outputs and gradients (for gradients, under a different assumption on the dimension of the matrix) remain constant through the different layers. The results presented differ from previous work in that they give nice concentration bounds for such output and gradient norms. In addition the authors prove results in the case of infinite samples under the assumption that they arise from a finite dimensional space.

Overall the presentation is clean, but the results presented have the following issues.

1.Very similar in nature to previous results, for example the results presented in [1] Theorem 5.4 give very similar concentration results and use very similar mechanisms.

2. All the results for the infinite stream coming from a finite dimensional subspace do not address the fact that the training points usually do not lay in a linear subspace of dimension d << n. Further, there is a stringent requirement on d in such results and they fail to hold for even "fairly" small d.

3. In Theorem 2, the argument that the output through a layer of a rank-d linear subspace remains within a rank-d linear subspace does not seem correct, is it possible that it remains in the union of subsets each of which lies in a subspace of rank d?

4. The proofs for the gradients make assumptions that deviate from the initialization previously introduced. The work of Glorot et al. for example discusses the tradeoffs between the two assumptions and suggests the balancing between maintaining the input and output variance distributions.

Due to the following issues I choose to reject this work at this time.

Below are additional minor typos or issues in the paper.
On page 6, the sentence that starts with: "it is beneficial for the width": Suggesting to use neural networks of constant width seems a bit impractical.

The citation of Arpit et al. from 2017 seems possibly wrong since the paper cited discusses memorization and does not focus at the initialization of neural networks.


TYPOS:
page 1 last sentence before equation 1:  back backward?

page 1 last sentence: repeated words: as as the the

page 1 last sentence: property --> properties?

page 2 section 2 paragraph 1: both these papers --> both papers, both of these papers?

page 4 sentence after the proof sketch paragraph: the sentence is difficult to read

[1] "Stochastic gradient descent optimizes over-parameterized deep relu networks", 2018, Difan Zou, Yuan Cao, Dongruo Zhou, Quanquan Gu




**Experience Assessment:**

I have read many papers in this area.

**Review Assessment: Checking Correctness Of Derivations And Theory:**

I assessed the sensibility of the derivations and theory.

**Review Assessment: Checking Correctness Of Experiments:**

I assessed the sensibility of the experiments.

**Review Assessment: Thoroughness In Paper Reading:**

I read the paper thoroughly.

---

### Decision · Program_Chairs · 2019-12-19

**Decision:**

Reject

**Comment:**

The article studies benefits of over-parametrization and theoretical properties at initialization in ReLU networks. The reviewers raised concerns about the work being very close to previous works and also about the validity of some assumptions and derivations. Nonetheless, some reviewers mentioned that the analysis might be a starting point in understanding other phenomena and made some suggestions. However, the authors did not provide a rebuttal nor a revision.